# Loss of *calpain3b* in Zebrafish, a Model of Limb-Girdle Muscular Dystrophy, Increases Susceptibility to Muscle Defects Due to Elevated Muscle Activity

**DOI:** 10.3390/genes14020492

**Published:** 2023-02-15

**Authors:** Sergey V. Prykhozhij, Lucia Caceres, Kevin Ban, Anna Cordeiro-Santanach, Kanneboyina Nagaraju, Eric P. Hoffman, Jason N. Berman

**Affiliations:** 1Children’s Hospital of Eastern Ontario (CHEO) Research Institute & University of Ottawa, Ottawa, ON K1H 8L1, Canada; 2Department of Psychology & Neuroscience, Dalhousie University, Halifax, NS B3H 4J1, Canada; 3AGADA BioSciences, Halifax, NS B3H 0A8, Canada; 4School of Pharmacy and Pharmaceutical Sciences, Binghamton University—State University of New York, Binghamton, NY 13902, USA

**Keywords:** limb-girdle muscle dystrophy, calpain 3, muscle, Duchenne muscular dystrophy, cholinesterase inhibitor, methylcellulose, birefringence, zebrafish, disease model

## Abstract

Limb-Girdle Muscular Dystrophy Type R1 (LGMDR1; formerly LGMD2A), characterized by progressive hip and shoulder muscle weakness, is caused by mutations in *CAPN3*. In zebrafish, *capn3b* mediates Def-dependent degradation of p53 in the liver and intestines. We show that *capn3b* is expressed in the muscle. To model LGMDR1 in zebrafish, we generated three deletion mutants in *capn3b* and a positive-control *dmd* mutant (Duchenne muscular dystrophy). Two partial deletion mutants showed transcript-level reduction, whereas the RNA-less mutant lacked *capn3b* mRNA. All *capn3b* homozygous mutants were developmentally-normal adult-viable animals. Mutants in *dmd* were homozygous-lethal. Bathing wild-type and *capn3b* mutants in 0.8% methylcellulose (MC) for 3 days beginning 2 days post-fertilization resulted in significantly pronounced (20–30%) birefringence-detectable muscle abnormalities in *capn3b* mutant embryos. Evans Blue staining for sarcolemma integrity loss was strongly positive in *dmd* homozygotes, negative in wild-type embryos, and negative in MC-treated *capn3b* mutants, suggesting membrane instability is not a primary muscle pathology determinant. Increased birefringence-detected muscle abnormalities in *capn3b* mutants compared to wild-type animals were observed following induced hypertonia by exposure to cholinesterase inhibitor, azinphos-methyl, reinforcing the MC results. These mutant fish represent a novel tractable model for studying the mechanisms underlying muscle repair and remodeling, and as a preclinical tool for whole-animal therapeutics and behavioral screening in LGMDR1.

## 1. Introduction

*CAPN3* is a gene encoding Calpain 3 Ca^2+^-dependent protease with autoproteolytic activity and re-activation capacity, expressed in muscle cells and involved in muscle function through proteolytic and localization/scaffolding mechanisms [1]. *CAPN3* was isolated by positional cloning based on the genomic samples of limb-girdle muscular dystrophy (LGMD) patients and selection for muscle-expressed genes in the identified genomic interval [2], thus revealing a genetic determinant of LGMD for the first time. A large body of clinical, biochemical, cell biological, and genetic studies has improved our understanding of the roles played by CAPN3. LGMD type R1 is currently linked to *CAPN3* mutations distributed over the length of the entire CAPN3 protein, most of which are recessive, but some are dominant, suggesting that the patients may benefit from gene therapy approaches, some of which are currently undergoing clinical trials [3]. Early biochemical studies of CAPN3 mutants revealed that they typically lose the proteolytic activity on their substrates but not always their autolytic activity or titin binding [4]. An integrated biochemical-cell localization study showed that autolysis of CAPN3 occurs intra- and intermolecularly and is required for protease activation of the inactive CAPN3 normally bound to titin at the N2A line of the sarcomere [5]. Upon activation, CAPN3 cleaves talin, filamins A and C, vinexin, ezrin, and titin Z-disk and M-line domains and co-localizes with these proteins at the Z-disk and M-lines. Interaction of CAPN3 with titin is particularly important because CAPN3 is inactive when bound to titin and, according to one model, the dispersal of CAPN3 enables its rapid activation upon molecular injury [6]. Ca^2+^ level increases and decreases are essential for muscle contraction and relaxation, respectively, which requires tight regulation of Ca^2+^ homeostasis [7]. Recent research has implicated decreases in sarco/endoplasmatic Ca^2+^-ATPase (SERCA), RyR1, and CaMKII levels under *CAPN3*-deficient conditions into Ca^2+^ dysregulation (more persistent elevated calcium levels), which can promote mitochondrial damage, endoplasmic reticulum stress response, oxidative stress, and apoptosis, also frequently observed in CAPN3-deficient muscles [3,7,8].

*Capn3* knockout mice were generated several years after CAPN3 was linked to LGMD and provided unique and highly relevant insights [9,10]. The first *Capn3* knockouts were, once born, viable and fertile, but the homozygous mutants did not survive gestation with expected frequency; the mutants also experienced mild muscular dystrophy, which could be linked to increased sarcolemmal permeability detected using Evans Blue dye labeling [9]. A later study demonstrated a mild atrophic phenotype of *Capn3* mutants with smaller muscle cross-sections, signs of inflammation in the muscles, and misalignment of sarcomeres [10]. The same group also compared wild-type and *capn3* knockout mice under either standard conditions or after induced endurance exercise testing using transcriptomic methods, revealing a perturbed muscle adaptation transcriptional response in the mutants, including reduced myofibrillar, mitochondrial, oxidative lipid metabolism, and fatty acid metabolism genes due to decreased PGC1α [11].

In zebrafish, there are two *CAPN3* homologs: *capn3a* and *capn3b*. Expression of *capn3a* is limited to the eye lens (from 2 dpf) and the brain (at the 4 dpf larval stage) [12], whereas *capn3b* was initially studied in the context of intestine and liver and only recently was detected in zebrafish muscle, as an incidental observation [13]. Capn3b was initially identified in zebrafish as a protease interacting with digestive organ expansion factor (Def) to control the levels of p53 protein in digestive organs by direct proteolysis [12]. Loss of *def* was earlier demonstrated to result in p53 activation and cell-cycle arrest, resulting in perturbed development of digestive organs [14]. Capn3b-Def interaction to control p53 degradation was then analyzed in greater detail, including Def post-translational modification and microscopy studies of nucleolar co-localization of these proteins [15]. Most recently, *capn3b* null small-deletion mutant zebrafish helped identify the roles of *capn3b* in liver development and regeneration, including under heat-stress conditions, to which these mutants responded with a striking curved-body phenotype [13].

In the current paper, we genetically inactivated *capn3b* gene in zebrafish to study its mutant phenotype as a potential LGMD disease model. We generated partial deletions and a null RNA-less allele, which allowed us to determine that *capn3b* mutations are not homozygous-lethal, but have significant subtle phenotypes. The RNA-less mutants were previously shown to produce loss-of-function phenotypes without the downside of genetic compensation [16]. Additional challenges to muscle function in wild-type and *capn3b* mutant zebrafish allowed us to identify dramatic differences in muscle damage susceptibilities under ‘challenge’ conditions. This short report primes the LGMDR1 zebrafish disease model for further research with regard to functional studies, transcriptomic response analyses, and drug screening to identify potential ameliorating agents.

## 2. Materials and Methods

### 2.1. Zebrafish Husbandry

Zebrafish experiments and husbandry follow standard protocols [17] in accordance with the standards of the University of Ottawa Animal Care Committee (approval number: CHEOe-3242-R2). All the work described here has been performed in the *casper* compound pigment mutant strain [18]. Zebrafish embryos were maintained at 28.5 °C during development and as adults. Embryos were grown in 1× E3 medium (5 mM NaCl, 0.17 mM KCl, 0.33 mM CaCl_2_, 0.33 mM MgSO_4_).

### 2.2. In Vitro Transcription for mRNA Expression

Plasmid DNAs for mRNA expression vectors were linearized with NotI overnight, extracted with Phenol:Chloroform:Isoamyl Alcohol (25:24:1) (Thermo Fisher, 15593031, Waltham, MA, USA) in Phase Lock Light 1.5 mL tubes, and precipitated with ethanol and sodium acetate, with added glycogen [19]. Cas9 mRNA was made from pT3TS-nCas9n plasmid [20] (Addgene, 46757, Watertown, MA, USA) after its linearization with XbaI and purifications as above. mRNAs were then synthesized using a mMESSAGE mMACHINE^®^ T3 Transcription Kit (Thermo Fisher Scientific, AM1348, Waltham, MA, USA) in 20 µL reactions and purified by LiCl precipitation according to the manufacturer’s instructions.

### 2.3. Generation of capn3b and dmd Mutants

The *capn3b* gene multi-exon deletions were performed using 6 single guide RNA (sgRNA) (3 targeting exon 4 and 3 targeting exon 6), designed using a combination of SSC (http://cistrome.org/SSC/) (accessed on 5 December 2018) [21] and CCTop (https://cctop.cos.uni-heidelberg.de:8043/index.html) (accessed on 5 December 2018) [22] websites. For the promoter and first exon deletion strategy (RNA-less mutant), 6 sgRNAs were designed similarly, 3 of which targeted upstream of the proximal promoter and 3 inside the first intron. The *dmd* mutant was likewise generated using 6 sgRNAs, 3 each for exons 24 and 25. Oligos containing T7 promoter, sgRNA spacer, and a scaffold overlap region were synthesized (Appendix A). sgRNAs were generated by performing an overlap-extension PCR of the sense sgRNA oligos, each combined with *Rev_sgRNA_scaffold* oligo (Appendix A). sgRNA templates were synthesized using *Taq* DNA polymerase (ABM, G009) by combining 10 μL of 10× buffer, 6 μL of 25 mM MgSO_4_, 2 μL of 10 mM dNTP, 5 μL of each oligo at 25 μM, 71 μL water, and 1.5 μL of Taq according to the following program: 94 °C for 5 min; 5 cycles: 94 °C for 30 s, 55 °C for 30 s, 72 °C for 30 s. The resulting PCR products were purified using a QIAGEN Gel Extraction kit (QIAGEN, 28704) and used for in vitro transcription using MEGAshortscript T7 kit (Thermo Fisher Scientific, AM1354). The sgRNAs were purified according to the kit instructions. Eggs were injected with Cas9 mRNA (250 ng/µL) and a mix of sgRNAs (300 ng/µL). The genotyping of both embryos and adults was performed by first preparing DNA samples for PCR using NaOH and Tris buffer, as described previously [23]. The PCRs were then run with *capn3b_GT-del*, *capn3b_rnaless_assay*, or *dmd_del* primers (Appendix A) using *Taq* DNA polymerase according to the touch-down PCR method: 94 °C for 3 min; 10 cycles: 94 °C for 30 s, 61 °C (with 1 °C decrease every cycle), 72 °C for 30 s, 25 cycles: 94 °C for 30 s, *51* °C, 72 °C for 30 s. The resulting PCR products for the wild-type and mutant alleles were visualized by standard agarose gel electrophoresis.

### 2.4. RNA Extraction and cDNA Synthesis

RNA was purified either from the single embryos of various genotypes at 48 hpf using the Direct-zol RNA Microprep kit (Zymo Research, R2061, Irvine, CA, USA) or was extracted from 30–50 zebrafish embryos using 500 µL Trizol reagent (Thermo Fisher Scientific, 15596026) and purified according to the Phasemaker Tubes protocol (Thermo Fisher Scientific, A33248). For cDNA synthesis, a 4-µg aliquot of total RNA was treated with TurboDNAse using a TurboDNA-free kit (Thermo Fisher Scientific, AM1907). cDNA was produced using a LunaScript RT Supermix (5×) kit in 10 μL reactions, according to the kit instructions (NEB, E3010L, Ipswich, MA, USA).

### 2.5. Reverse Transcription-Polymerase Chain Reaction (RT-PCR) Analysis

To analyze mutation consequences on the corresponding transcripts, we used Q5 High-Fidelity 2X Master Mix (NEB, M0492S) to amplify several regions in *capn3b* transcripts with *capn3b_cDNA-del_assay*, *capn3b_fullCDNA*, and *capn3b_RLtest* primer pairs, as well as a *dmd_cDNA* assay (Appendix A) in the *dmd* cDNA according to the standard Q5 program with a 65 °C annealing temperature.

### 2.6. Sanger Sequencing and Analysis

All of the deletion and wild-type PCR products were Sanger-sequenced with 2–6 replicates using the primers to amplify them from both directions at the Ottawa Hospital Research Institute StemCore laboratories. The sequencing reads were aligned to either their genomic or transcriptomic references using Vector NTI 10 software version 10.3.0 (Invitrogen), with manual splitting of sequences at the breakpoints and the alignments exported as pdf files for documentation and figure generation. Deletion sizes were quantified based on the alignments.

### 2.7. In Situ Hybridization

The template for the *capn3b* anti-sense RNA probe was amplified with *capn3b_ISH_for* and *T7_capn3b_ISH_rev* primers (Appendix A) from the pooled wild-type embryo cDNA using Q5 2x Master Mix at 66 °C annealing temperature. The probe was synthesized from the template PCR product using a DIG RNA Labeling Kit (SP6/T7) (Roche, 11175025910, Basel, Switzerland) according to the kit instructions. Whole-mount in situ hybridization (WMISH) was carried out according to the protocol by Lauter et al. [24], except that the detection step was performed using Anti-Digoxigenin-AP, Fab fragments (Roche, 11093274910) at 1:2500 dilution in the blocking buffer, and the staining step was performed with a BCIP/NBT Alkaline Phosphatase (AP) Substrate Kit (Vector Laboratories, SK-5400) according to the kit instructions. The stained embryos were then fixed in 4% PFA for 30 min, washed in PBST, and embedded into 80% glycerol for imaging.

### 2.8. Methylcellulose Preparation and Treatment

Methylcellulose (MC) solutions are used to create more challenging movement conditions for zebrafish embryos and larvae and thus identify genetic conditions sensitive to such environments. To prepare 400 mL preparation of 0.8% MC (MilliporeSigma, M0387, Burlington, MA, USA) in E3 medium, 260 mL of E3 medium were chilled at −20 °C for 30 min and placed on ice. Separately, 140 mL of E3 medium were heated to 80 °C and 3.2 g of MC were added. The hot suspension was agitated with a magnet stir bar until all particles were wetted and evenly dispersed. The ice-cold E3 was added to the MC suspension in a covered glass beaker and then moved to the 4 °C fridge overnight and stored for up to a week.

To start the treatments, we pre-warmed the necessary volume of MC medium to room temperature in a water bath. Embryos at 48 hpf were treated in 1 mL of E3 medium with 50 uL of 10 mg/mL pronase stock solution for 15–20 min, washed by several rinses, and returned to a weakly-tricained medium (500 µL of 0.4% Tricaine per 40 mL of E3). The embryos were collected in approximately 1 mL of E3, washed of tricaine, and moved to control or 0.8% MC E3 medium. The media was changed daily, and the final analysis was performed at 5 dpf by birefringence.

### 2.9. Evans Blue Dye Assay

Evans Blue Dye (EBD) assay was performed precisely according to a published protocol [25] and then imaged using Zeiss Axio Observer. Evans Blue Dye assay had to be performed at 4 dpf or earlier because injections into the Common Cardinal Vein at 5 dpf were largely ineffective and many affected MC-treated embryos had some degree of heart edema.

### 2.10. Azinphos-Methyl Treatment

Azinphos-methyl (APM) (MilliporeSigma, 45333) was reconstituted to 60 mM in Dimethyl Sulfoxide (DMSO); diluted further in DMSO to 6, 10, or 15 mM; and then diluted 1:20,000 to its final concentrations of 0.3, 0.5, or 0.75 µM. DMSO was used as a vehicle control (2 µL per 40 mL of the medium). These concentrations were determined empirically by testing the differential response of wild-type and *capn3b* mutant embryos to multiple APM concentrations.

### 2.11. Birefringence Analysis

Muscle structure was assessed on a ZEISS Axio Zoom V16 microscope using Analyzer S, rotatable, d = 66 mm (Zeiss, 435530-0000-000, Jena, Germany) to detect birefringence signals, which allows visualization of the overall structure of the muscle and many of its large-scale abnormalities.

### 2.12. Statistical Analysis and Visualization of the Data

The significances of the phenotype proportion differences after MC or APM treatment were determined by the Cochran–Mantel–Haenszel test using R packages ‘psych’, ‘stats’, and ‘rcompanion’. In particular, ‘mantelhaen.test’, ‘groupwiseCMH’ functions were used. The graphs were made using the ‘ggplot2’ package. Code, and data used in this study are available in the following repository: https://github.com/SergeyPry/capn3b_paper.

## 3. Results

### 3.1. Both RNA-Less and Partial Deletion capn3b Mutants Disrupt capn3b Expression, but the Mutants Have Normal Muscle Morpholog and Are Viable and Fertile

Given that *CAPN3* mutations are a frequent cause of limb-girdle muscular dystrophy type R1 (LGMDR1) in humans [3], we sought to generate zebrafish mutants of CAPN3 orthologs in zebrafish. Because *capn3a* is expressed in the eye lens and not in muscles, we targeted the *capn3b* gene, for which there is evidence for tail musculature expression [13] and aortic arch smooth muscle staining [26] in zebrafish. However, the *capn3b* gene has not been studied in zebrafish in the muscle context until now. We used Clustered Regularly Interspaced Short Palindromic Repeats (CRISPR)/Cas9 strategies to generate an RNA-less (proximal promoter and exon 1) deletion mutant and a deletion mutant from exon 3 to 5 (Figure 1A). Both strategies were successful and, after the necessary breeding, the resulting homozygous mutants were verified by standard PCR genotyping and sequencing of the genotyping amplicons in wild-type and mutant embryos. The *rnaless* mutant allele was a 751-bp deletion, and the *mut1* and *mut73* alleles were a 33-bp insertion with a 467-bp deletion and a 608-bp deletion, respectively (Figure 1B). We then verified *capn3b* mRNA presence, size, and sequence in the wild-type and mutant embryos with three cDNA-level assays (Figure 1C). In order to confirm that the mutant transcripts or lack thereof correlate with the mRNA species predicted by genomic sequence deletion analysis, we designed three types of cDNA assays: a cDNA-del assay, a Rnaless-test assay, and a full-cDNA assay, which are RT-PCR assays with specific pairs of primers (Figure 1C). Homozygous *capn3b rnaless* mutants were negative for all of the cDNA assays, thus confirming the ability of this mutation to fully inactivate *capn3b* transcription (Figure 1C). cDNA-del assay used to verify the multi-exon deletion mutants indeed confirmed that *mut1* and *mut73* alleles result in smaller cDNA PCR products, whereas the RNA-less test assay resulted in comparable PCR products in all samples except the rnaless homozygotes (Figure 1C). Amplification of the whole *capn3b* cDNA with the full-cDNA assay in the wild-type and all *capn3b* mutants allowed determination of the exact translation consequences in *mut1* and *mut73* mutants, which both have a transcript with an in-frame 164-codon deletion, and *mut1* homozygotes also have a transcript coding for a dramatically truncated protein (Figure 1C,D). These results are significantly different from the predictions made based on the genomic sequences from the mutants, which only suggested truncated proteins. The in-frame deleted transcript can be explained by the skipping of exon 2 and the deletion-generated hybrid exon 3/5, whereas the truncated protein transcript is due to the expected splicing. We then analyzed expression of *capn3b* by WMISH in the wild-type, *capn3b*^−/−^ (*rnaless*), *capn3b*^mut1/mut1^, and *capn3b*^mut73/mut73^ embryos at 28 and 52 hpf and showed clear tail muscle expression of *capn3b* in the wild-type embryos, completely absent expression of *capn3b* in *capn3b*^−/−^, and dramatically reduced expression in *capn3b*^mut1/mut1^ and *capn3b*^mut73/mut73^ embryos (Figure 2A). In a separate and subsequent experiment, we stained *capn3b* mRNA in 3 dpf wild-type and *capn3b*^−/−^ embryos, which confirmed *capn3b* expression in the tail musculature and branchial arch regions (Figure 2B). To assess whether lack of *capn3b* affects the development of particular muscle groups, we also stained wild-type and *capn3b*^−/−^ embryos for the *unc45b* myosin chaperone gene expressed universally in the muscle [27] and showed that both the tail and ventral head musculature *unc45b* staining is normal in *capn3b*^−/−^ embryos (Figure 2B). Adult *capn3b* mutants were also morphologically normal, viable, and fertile. As a result, we focused on the muscular phenotypes in larvae under challenge and did not pursue detailed observations and measurements of adults.

### 3.2. A Mutant in dmd Shows a Strong Muscular Dystrophy, Thus Serving as a Positive Control

As part of a larger project on muscle dystrophy models, we also generated a mutant in the dystrophin gene *dmd* by targeting exon 24 and 25 with two sgRNA (Figure 3A). This effort resulted in a 767-bp deletion in the genomic sequence (Figure 3B). The corresponding transcript-level deletion can be visualized by PCR (Figure 3C) and was sequenced to be a 155-bp deletion (Figure 3D). In this case, two *dmd* exon fragments were joined and no exon skipping occurred. The translation of the mutant transcript is expected to result in a truncated protein with 70% of its length missing (Figure 3E). Homozygous *dmd*^–/–^ mutants were obtained from *dmd*^+/−^ crosses in the expected proportion (~25%) and exhibited the expected muscular dystrophy phenotype under birefringence analysis (Figure 3F). We used this phenotype as a positive muscular dystrophy control to calibrate the phenotypes of other zebrafish mutants we have generated with more subtle muscular dystrophy phenotypes, such as *capn3b* mutants.

### 3.3. Methylcellulose Incubation Reveals Increased Susceptibility of capn3b Mutants to Muscle Damage

To identify any possible phenotypes of *capn3b* mutants, we decided to apply challenge techniques to the skeletal muscles by incubating 2 dpf zebrafish embryos in a viscous solution. After several trials, we settled on 0.8% methylcellulose (MC) (1500 cP) as the viscous medium of choice, which does not induce high levels of lethality but is still potent enough to produce phenotypes after 2 or 3 days of incubation (2–4 or 2–5 dpf). After initial trials, we selected *wild-type*, *capn3b*^−/−^ (*rnaless*), and *capn3b*^mut1/mut1^ as relevant study systems because *capn3b*^mut73/mut73^ embryos behaved almost identically to the wild-type, likely because their sole *capn3b* transcript has an in-frame deletion. Under control conditions after 2–5 dpf incubation in fish medium, almost all *wild-type*, *capn3b*^−/−^, and *capn3b*^mut1/mut1^ had indistinguishable normal muscle structure, except for approximately 1% (4 of 485) of wild-type and 4% (20 of 467 and 22 of 495) of *capn3b*^mut1/mut1^ and *capn3b*^−/−^ embryos, which exhibited abnormal muscle structure (Figure 4). By contrast, 2–5 dpf incubation in 0.8% MC increased the fraction of wild-type embryos with abnormal muscle structure to 4% (*p*-value = 0.01) and dramatically elevated the fraction of muscle abnormalities in *capn3b*^mut1/mut1^ and *capn3b*^−/−^ embryos to approximately 30% (151 of 480 (*p*-value = 1.22 × 10^−26^) and 131 of 456 (*p*-value = 8 × 10^−26^), respectively) (Figure 4). This result suggests a much greater susceptibility of *capn3b* mutants to muscle damage under challenge conditions where greater muscle activity is required.

### 3.4. No Evidence of Sarcolemmal Damage in capn3b Mutants but Disrupted Permeability in dmd Mutant Embryos

Evans Blue Dye (EBD) staining is an established approach to visualize instances of sarcolemmal damage in multiple animal models of muscular dystrophies [28]. In zebrafish, EBD is injected into the bloodstream, together with a 10-kDa fluorescent dextran solution and then, during incubation, zebrafish undergoing muscle damage will accumulate significant amounts of EBD, resulting in red fluorescence, whereas the undamaged embryos will contain the dye only in the bloodstream [25]. We first performed EBD assay in the progeny of *dmd*^+/−^ zebrafish at 3 dpf. This analysis revealed that the majority of *dmd*^–/–^ embryos at 3 dpf contained numerous labeled muscle fibers, whereas the siblings (*dmd*^+/−^ and *dmd*^+/+^) contained signal only in the bloodstream (Figure 5A).

For the *capn3b* embryos treated with MC, it was not practical to perform EBD assay at 5 dpf, the endpoint of the experiment, due to physical penetration issues and heart edema, so we performed this analysis at 4 dpf, when the phenotypes are milder, or when the fish looked normal. The MC-treated mutant embryos were pre-sorted under the birefringence filter to have a detectable degree of birefringence signal reduction. The EBD-dextran mix was injected into groups of ~50 embryos of wild-type, *capn3b* mut1/mut1, *capn3b* rnaless/rnaless grown either under control conditions or E3 medium with 0.8% methylcellulose. Successfully labeled larvae with bloodstream EBD and FITC-dextran signals were then imaged for EBD signal and FITC-dextran, which showed that none of the animals in any experimental group had muscle fiber labeling (Figure 5B). This result suggests that *capn3b* loss does not lead to sarcolemma damage after 2 days of MC treatment.

### 3.5. capn3b^−/−^ Embryos Are More Susceptible to Muscle Damage Than Wild-Type after Treatment with Low Concentrations of Azinphos-Methyl

Although we obtained very relevant results on the increased susceptibility of *capn3b* mutants to muscle damage using MC treatments, working with methylcellulose solutions is cumbersome, precluding higher throughput analysis. This prompted us to explore other options to stimulate excessive muscle activity. Azinphos-methyl (APM) is an inhibitor of Acetylcholinesterase (AChE), known to control cholinergic neurotransmission in animals but may also have other roles [29]. APM application in zebrafish results in muscle hyperactivation (myotonia) due to persistently high levels of acetylcholine, and consequent muscle damage is easily detectable by the birefringence assay [29,30]. We reasoned that starting treatment later at 2 dpf and using low micromolar and sub-micromolar APM concentrations can help identify treatments to which wild-type and *capn3b* mutant embryos can be differentially sensitive. For this experiment, we used *wild-type* and *capn3b*^−/−^ (*rnaless*) embryos and 0 (control medium with DMSO), 0.3, 0.5, and 0.75 µM APM treatments from 2 to 4 dpf (Figure 6A). Birefringence imaging and statistical analyses of the numbers of embryos with normal and abnormal muscles after APM treatments demonstrated that at 0.3 and 0.5 µM APM, there is a highly significant elevated susceptibility of *capn3b*^−/−^ embryos to muscle damage due to hyperactivation by AchE inhibition, whereas in the control media both types of embryos are normal, and at 0.75 µM APM they both have similar rates of abnormal muscles (Figure 6B). To confirm the activity of APM treatments, we performed WMISH for the *hspb11* cholinesterase inhibition marker on *wild-type* and *capn3b*^−/−^ embryos treated with 0.3 µM APM and found qualitatively similar levels of induction (Figure 6C). This result further confirms that *capn3b* loss results in increased susceptibility to damage after muscle hyperactivation.

## 4. Discussion

We first confirmed that *capn3b* is a *bona fide* muscle-specific gene, consistent with a previous study [13], and the major zebrafish *CAPN3* homolog expressed in the muscle because it is not known if *capn3a* is expressed at a low level in the muscle. Using two CRISPR/Cas9 mutagenesis methods, we produced three *capn3b* mutants: a null (RNA-less) mutant and two partial deletion mutants. Both partial deletion mutants (mut1 and mut73) produced a 164-codon in-frame deletion transcript, but *mut1* also had a frameshifted mRNA, whereas mut73 predominantly expressed the in-frame deletion transcript resulting from exon skipping and was resistant to nonsense-mediated decay [31]. Under normal growing conditions, neither the embryos nor adults of any of these mutants showed significant phenotypes associated with known zebrafish muscular dystrophy models, such as was observed in the *dmd* mutants we generated to serve as a positive control. By contrast, incubating wild-type and *capn3b* loss-of-function mutants (mut1 and RNA-less) in viscous methylcellulose-containing media revealed that the mutants are much more susceptible to muscle damage than the wild-type embryos, as visualized by abnormal birefringence signals in their muscles. However, we failed to identify a clear increase in membrane permeability after methylcellulose treatment in the mutants using the established EBD assay. Because there were technical limitations in our EBD assay experiments, further experiments using different bloodstream injection techniques and additional live imaging methods will help establish if Capn3b is involved in membrane integrity maintenance. On the other hand, Capn3 proteins are also responsible for multiple functions in the sarcomere and cytosol of muscle cells, and these could be sufficient to explain the CAPN3 deficiency phenotypes. Treatment of wild-type and *capn3b^−/−^* (rnaless) embryos with low sub-micromolar concentrations of a cholinesterase inhibitor to hyper-activate their muscles showed again that *capn3b* loss dramatically increases susceptibility to muscle damage. Given the general consensus that mutants of many genes in zebrafish do not have an easily discernible embryonic phenotype under steady state conditions [32,33], these results provide a valuable framework for investigating a phenotype among mutants in skeletal muscle-related genes, by challenging the mutants with the described or alternative means. These results may help disprove the hypothesis that *capn3b* is necessary for maintaining muscle structure and sarcolemma integrity under normal conditions and instead support the idea that calpain 3 function is required under challenging conditions.

This study is the first to address the skeletal muscle-specific function of *capn3b* in zebrafish. This study has not addressed whether skeletal muscle Capn3b interacts with any factors relevant for cell-cycle progression, as is the case for Def and p53 in the intestinal and liver cells. One possibility is that intestinal and liver cells have more extensive cell proliferation, and therefore specialized control over this process requiring a layer of regulation involving Capn3b or different modes of stress might exert a *capn3b* phenotype on these tissue types.

Zebrafish have become an important model for multiple types of muscular dystrophies, including LGMD [34]. However, direct comparison between ours and other zebrafish LGMD models is difficult, as many of them were generated using morpholinos [35,36,37]. We also obtained highly striking phenotypes using *capn3b* translation-blocking morpholinos, but they were not replicated by the mutant phenotypes, even in the case of our *capn3b* rnaless mutants. Thus, the more appropriate comparison would be mouse *Capn3* knock-out models, which are viable and fertile but still show some impairment in muscle functions. However, the zebrafish pectoral fins and limbs of tetrapod vertebrates share some basic developmental homology, but are highly divergent [38]. The physical stresses experienced by land-based vertebrates likely place more importance on CAPN3 in mammals, including humans, than for aquatic-based organisms, such as zebrafish. An important question in this regard remains whether or not Calpain 3 functions the same way in all muscle types and whether there is any redundancy among calpain family members expressed in the skeletal muscle. Challenge of muscle function in zebrafish using methylcellulose has been described in *cavin-1* and *caveolin-3* morphants and transgenics, respectively, where the importance of these genes for sarcolemmal integrity was shown [35] and for *bag3* morphants, where MC treatment resulted in myofibrillar disintegration [37]. However, the exact concentration, stages, and timing of treatment we independently optimized for the purposes of this study. Application of the APM acetylcholinesterase inhibitor was previously described to result in muscle damage and expression of muscle activity marker *hspb11* [29]. They also developed an *hspb11* promoter-based biosensor transgenic line to optimize the screening of similar compounds [30]. We used these studies as a starting point and optimized the treatment to reveal a differential response in wild-type vs. *capn3b* mutants. A related future plan is the application of low-dose APM treatment to other LGMD mutants without a strong phenotype. Similarly to the previous work in mice [11], we plan to perform muscle transcriptomic analyses of wild-type and *capn3b* mutants, with or without hyperactivation to identify more subtle effects.

In conclusion, we propose that studies of the non-essential genes required for muscle plasticity but not for organismal viability are important for uncovering the genes involved in physiological responses to different type of stressors. For example, *capn3b* seems to be required for skeletal muscle functions during either mechanical (MC-treatment) or functional/activational (APM-treatment) stress. This unique feature of genes with stress-induced mutant phenotypes can shed light on elucidating the genes involved in muscle plasticity under different types of physiological stressors.

## Figures and Tables

**Figure 1 genes-14-00492-f001:**
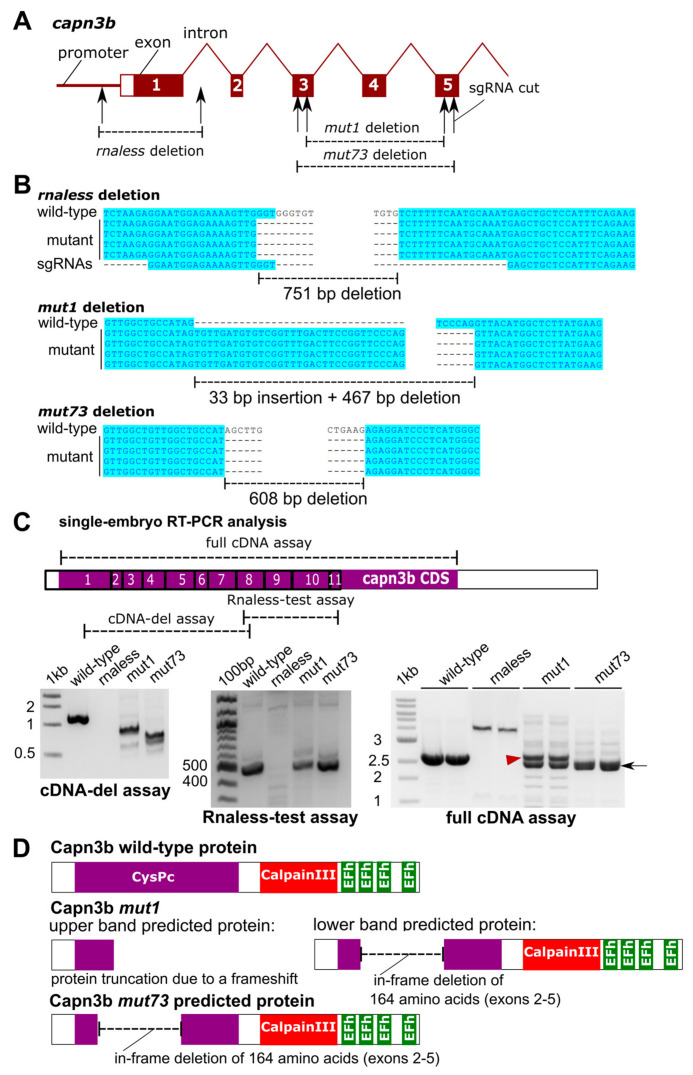
Generation and molecular characterization of the *capn3b* mutants. (**A**) Structure of the *capn3b* gene, including the promoter region, the first 5 exons, and introns. The targeted regions are shown where deletions were generated by the indicated sgRNAs. (**B**) Genomic sequencing of the deletion alleles and their alignment to the wild-type sequence. (**C**) Reverse transcription polymerase chain reaction (RT-PCR) analysis of *capn3b* transcripts in single-embryo cDNA samples of indicated genotypes. The regions of the *capn3b* mRNA analyzed by these assays are indicated in the diagram. Exon numbers are in the boxes. The bands in the mut1 and mut73 RT-PCR are indicated with a red arrowhead (upper band) and a black arrow (lower band). (**D**) Sequencing of the RT-PCR products of the full cDNA assay allowed protein-level interpretation of the identified mutations, as illustrated using protein domain diagrams.

**Figure 2 genes-14-00492-f002:**
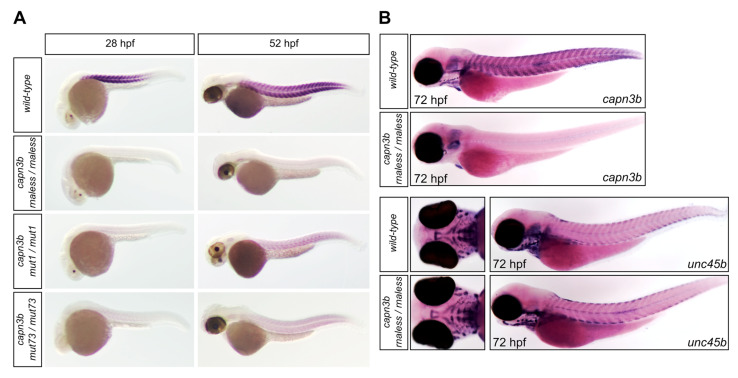
Expression pattern analysis of *capn3b* in wild-type and *capn3b* mutant zebrafish embryos. (**A**) Whole mount in situ hybridization with the *capn3b* probe, not including the deleted regions, was performed in *wild-type*, *capn3b* mut1/mut1, *capn3b* mut73/mut73, and *capn3b* rnaless/rnaless embryos at 28 and 52 hpf. (**B**) Whole mount in situ hybridization analysis with the *capn3b* and *unc45b* probes in *wild-type* and *capn3b* rnaless/rnaless embryos at 72 hpf. For each sample type, at least 40 embryos were stained. Representative images are shown in the figure (hpf = hours post-fertilization).

**Figure 3 genes-14-00492-f003:**
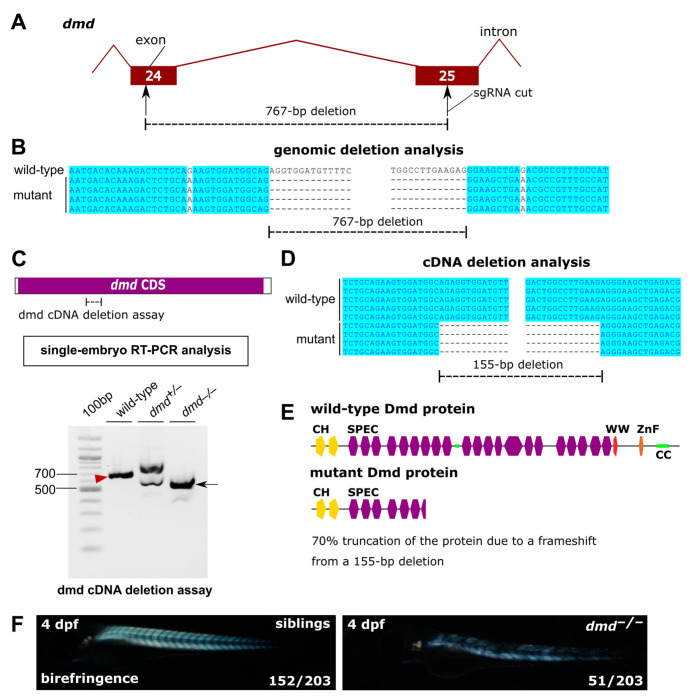
Generation and the phenotype of the zebrafish *dmd* mutant. (**A**) Structure of the targeted region of the *dmd* gene, including exons 24 and 25. The sgRNA cut positions are indicated. (**B**) Genomic sequencing of the deletion alleles and their alignment to the wild-type sequence. (**C**) RT-PCR analysis of the targeted *dmd* region. The *dmd* deletion assay region is indicated in the diagram. Results of *dmd* cDNA deletion assay in single embryo samples of the indicated genotypes are shown. The *dmd* cDNA deletion assay wild-type band is indicated with a red arrowhead and the mutant band is indicated with a black arrow. (**D**) cDNA deletion PCR product sequencing analysis shows the resulting 155-bp deletion at the cDNA-level. (**E**) Sequencing of the RT-PCR products of the *dmd* cDNA deletion assay allowed protein-level interpretation of the identified mutation, as illustrated using protein domain diagrams. (**F**) Birefringence analysis of *dmd*^+/−^ incross embryos shows approximately 25% of *dmd*^−/−^ embryos with disrupted muscle structure, whereas siblings show a normal birefringence pattern.

**Figure 4 genes-14-00492-f004:**
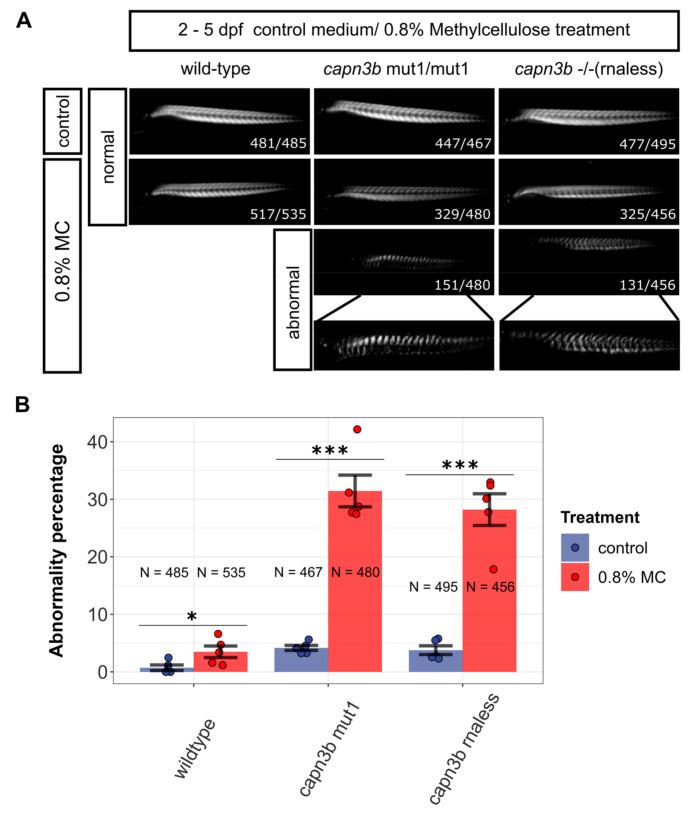
Mutants in *capn3b* gene are significantly more sensitive than wild-type zebrafish to growth in methylcellulose. (**A**) Birefringence imaging of *wild-type*, *capn3b*^mut1/mut1^, and *capn3b*^−/−^ (rnaless) mutant embryos after culturing them from 2 days post fertilization (dpf) to 5 dpf under control conditions or in 0.8% methylcellulose (MC) in E3 medium. Photographs of embryos with normal muscle morphology or disrupted (abnormal) are shown, except for wild-type embryos, which were almost completely resistant to the treatment. The combined numbers out of the total analyzed are shown at the lower right-hand corner of the representative images. (**B**) Plot of the percentages of embryos with abnormal muscle structure (abnormality percentage) for embryos of different genotypes. Significance is calculated using the Cochran–Mantel–Haenszel test using all available counts in 3-dimensional contingency tables. “***” indicates *p*-values < 0.001 and ‘*’—*p*-value < 0.05.

**Figure 5 genes-14-00492-f005:**
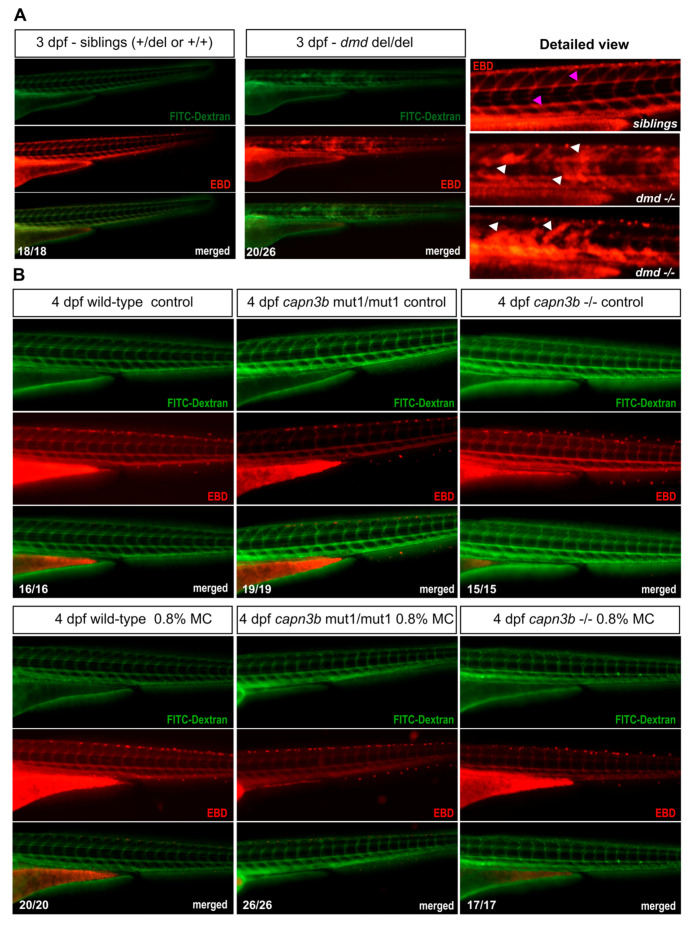
Evans Blue Dye analysis shows lack of permeability disruption by methylcellulose incubation in both wild-type and *capn3b* mutant embryos. (**A**) Evans Blue Dye (EBD) and FITC-Dextran injected sibling and *dmd^−/−^* embryos at 3 dpf. EBD signal is labeled in red and FITC-dextran in green. Merged images suggest a near complete co-localization of EBD and FITC-dextran. Vascular labeling indicates the normal situation where muscle permeability is not disrupted, whereas the more widespread labeling indicates disruption of muscle cell permeability. The detailed magnified view shows examples of sibling and *dmd^−/−^* embryo EBD labeling. White arrowheads show damaged muscle labeling and fuchsia arrowheads label EBD in the blood vessels in the sibling embryos. (**B**) EBD assay of *wild-type*, *capn3b*^mut1/mut1^, and *capn3b*^−/−^ mutant embryos after culturing them from 2 days post-fertilization (dpf) to 4 dpf under control conditions or in 0.8% methylcellulose (MC) in E3 medium. Only vascular labeling could be detected in all of the imaged embryos. The combined numbers out of the total analyzed are shown in the bottom right-hand corner of the representative images.

**Figure 6 genes-14-00492-f006:**
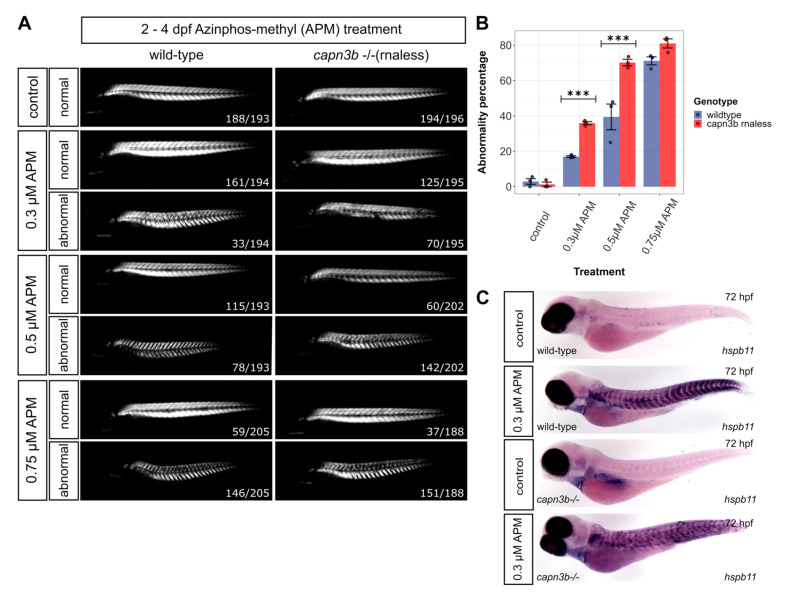
Elevated sensitivity of *capn3b^−/−^* (rnaless) mutant to cholinesterase inhibitor treatment. (**A**) Birefringence analysis of *wild-type* and *capn3b*^−/−^ (rnaless) mutant embryos after the control treatment or with increasing concentrations (0.3, 0.5, and 0.75 µM) of azinphos-methyl (APM). Images of embryos with normal and abnormal tail musculature for each genotype. The combined numbers out of the total analyzed are shown in the lower right-hand corner of the representative images. (**B**) Plot of the percentages of embryos of different genotypes with abnormal muscle structure (abnormality percentage) for different treatments. Significance is calculated using the Cochran–Mantel–Haenszel test using all available counts in three-dimensional contingency tables. “***” indicates *p*-values < 0.001. (**C**) Whole mount in situ hybridization for *hspb11* muscle activity marker in control and 0.3 µM APM-treated *wild-type* and *capn3b*^−/−^ (rnaless) mutant embryos from 2 to 3 dpf. Groups of 40 embryos were treated and analyzed, resulting in near-uniform labeling patterns shown in representative images (dpf = days post-fertilization).

## Data Availability

The data presented in this study are openly available in https://github.com/SergeyPry/capn3b_paper.

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
