# Peer review of "Loss of calpain3b in Zebrafish, a Model of Limb-Girdle Muscular Dystrophy, Increases Susceptibility to Muscle Defects Due to Elevated Muscle Activity"

_genes, 2023, doi:10.3390/genes14020492_

Round 1

Reviewer 1 Report

In this manuscript, Prykhozhij and colleagues focused on CAPN3, one of the significant responsible genes for LGMD. They showed that zebrafish homolog capn3b is expressed in larval muscle and tried to assay its role in the skeletal muscle by generating the loss of function mutants. Although the mutants did not show any significant phenotype in the normal condition, the authors found that those mutants are susceptible to the perturbation causing muscle damage, and suggested that those mutants and the perturbation system are the potentials to be a new model for studying the LGMD mechanism.   

The data are potentially novel and informative for the researchers employing the zebrafish as a disease model. However, there is a criticism with the paper that should be clarified prior. 

This reviewer’s concern is about genetic characterization of the mut1 and mut73. Regarding mut1, the authors detected two transcripts with the cDNA assay, although the detected genomic DNA deletion is a single pattern (33bp ins + 467bp del). The authors describe as below (line241-245)

 “Amplification of the whole capn3b cDNA with the full-cDNA assay in the wild-type and all capn3b mutants allowed determination of the exact translation consequences in mut1 and mut73 mutants, which both have a transcript with an in-frame 164-codon deletion and mut1 homozygotes also have a transcript coding for a dramatically truncated protein (Fig. 1C, D). These results are significantly different from the predictions made based on the genomic sequences from the mutants, which only suggested truncated proteins. (line245)"

If this is true, why and how does mut1 genomic DNA seq generate a transcript with an in-frame 164 codon? Furthermore, why and how mut73 genomic DNA seq not generate a transcript with a dramatically truncated protein? 

The authors should provide convincing explanations and discussion to those questions.

Without clarifying those points relating to genetic characterization of the mutations, I think mut1 and mut73 should not be used in the following assays for muscular damage phenotype. At least, mut1 should not be treated as same as a rna-less mutant.

On the other hand, the reviewer accepts the data with RNA-less mutant and the value of assay under the challenge condition with a rna-less mutant.

Reviewer 2 Report

The manuscript by Prykhozhij et al. describes their work to examine the role of capn3b in muscle biology in zebrafish.  CAPN3 has been linked to limb-girdle muscular dystrophy in patients.  In zebrafish there are two paralogs capn3a and capn3b.  Based on expression data, capn3a is not expected to contribute to muscle phenotypes, whereas capn3b likely would.  The authors have carefully crafted deletion mutants including a mutant that no longer produces detectable mRNA.  The latter is important is it should prevent some mechanisms of genomic compensation.  The authors did not detect phenotypes in these mutants in either larval or adult stages with standard rearing conditions.  However, evaluation under challenge conditions that required greater muscle activity.  The data and their presentation appear sound.  However, there are several aspects that I believe could improve the manuscript.

Major Concerns/Questions:

·      For EBD assay results, do you know if any of these stained animals that did not show staining did indeed have birefringence phenotypes?  Since this was done at 4DPF, it sounds like the birefringence phenotype was less penetrant (the levels are not provided), so theoretically possible that with numbers presented it could have been missed.  Can you confirm there were phenotypic fish in your analysis or provide phenotype penetrance at 4dpf?

·      In Figure 6C, do the APM treated fish shown represent those that show a birefringence phenotype?  Do other fish look like the control fish in these ISHs and do the numbers reflect what is observed in Fig 6A?  Can you confirm that these phenotypes were correlated in across individual fish or provide numbers for the ISH observations.  It is possible that the ISH phenotype is more sensitive/penetrant, which would be interesting to know.

Minor concerns/suggestions:

·      The rationale for creating the RNA-less mutant, which I presumed was to avoid genetic compensation was not mentioned, I would add a small note and reference.

·      The writing often contains overly complex or run-on sentences that complicate the authors’ message.  This is especially apparent in the introduction.  Some examples include the first sentence (lines 34-36), line 47-51, even line 52-55 would seem clearer as two sentences, line 253-257.  These are just some examples, but anything to clarify messaging will be helpful.

·      For the last sentence of the first paragraph of the introduction (lines 55-59) provide more context to the recent research and how it sheds light on capn3 function.  Right now, it is a list without being tied in well.

·      With all work in Casper fish there is a possibility that results may be skewed a bit due to inbreeding for line maintenance (typically), not quite like a mouse strain.  I do not think there is anything to do about this, but there was no rational provided for doing work in this background- so generically see this is a weakness.

·      In line 190, it is stated that “due to the physical feasibility reasons”, can you simply state the reason that makes it less feasible to do this at 5dpf vs 4dpf?  Similarly, line 367 also states it is not practical, but you could easily provide rationale.

·      In lines 223-226.  This sentence is difficult to follow though I understand what you are trying to do, I think this can be cleaned up.

·      Throughout the paper the mutant fish you referred to have a few different nomenclatures.  I recommend using official nomenclature from zfin guidelines.  For example, the rna-less mutant capn3bxx101/xx101.  You may have to determine your institution’s allele designator and next available alleles, but it will help with clarity and dissemination of information and eventual fish lines.

·      For lines 234-235, I would restate that you developed PCR assays to examine cDNA to examine expected alleles without naming the assay, but this is more my preference.

·      For line 245-246, perhaps give a better indication of how the results differ from the predicted.

·      In figure 1A, indicate that the gene goes beyond exon 5 with intron like Figure 3.

·      Line 257-259, the conclusion to focus on larvae since adults were morphologically normal makes it seem that there was a larval phenotype even though they were normal.  So, consider focusing more on “pursuing muscular phenotypes in larvae under challenge”

·      For Figure 1C and 3C it is difficult to translate your PCR fragments to the target deletions, adding exon boundaries to these cDNAs may help, or coloring the exons of importance from 1A and 3A.  This to a lesser degree impacts the interpretation of Figure 1D.

·      In Figure 3C, why is the non-mutant band in heterozygotes larger than the wild-type band?

·      Dmd alleles should use appropriate allele designators too (not del/del).

·      It would be better to list the N numbers per condition on Figure 4B and 6B

·      The first line of the Discussion should be changed, capn3b expression was noted in muscle by Chen et al. 2020 (pubmed #332588143).  Also, capn3a has not been detected in muscle by whole-mount in situ, but it could be present in low levels.  Might be worth cutting trunk/tails off larval fish and confirm by RT-PCR (not required for this manuscript, can simply soften statement a bit).

·      After line 425, can you add any thoughts as to why no increased membrane permeablity was observed or how this impacts understanding of mechanism.

·      There is a period before Zebrafish on line 444.

Round 2

Reviewer 1 Report

I am satisfied with the revisions that the authors have made.

With additional information, including revisions in response to another reviewer's comments (e.g., Fig. 1c), the structure of the mut1- and mut7 transcript became clearer. In addition, the information helps the readers to understand that unexpected transcript may be derived from exon skipping and that mut1 zebrafish likely have less of an in-frame deletion transcript.